# Oral Health and Risk of Retinal Vascular Occlusions: A Nationwide Cohort Study

**DOI:** 10.3390/jpm13010121

**Published:** 2023-01-05

**Authors:** Yoonkyung Chang, Sung-Hee Kim, Jimin Jeon, Tae-Jin Song, Jinkwon Kim

**Affiliations:** 1Department of Neurology, Mokdong Hospital, Ewha Womans University College of Medicine, Seoul 07985, Republic of Korea; 2Department of Neurology, Yongin Severance Hospital, Yonsei University College of Medicine, Yongin-si 16995, Republic of Korea; 3Department of Neurology, Seoul Hospital, Ewha Womans University College of Medicine, Seoul 07804, Republic of Korea

**Keywords:** periodontitis, tooth brushing, oral health, retinal vascular occlusion

## Abstract

Retinal vascular occlusions are a common cause of visual loss. The association between oral health and the risk of retinal vascular occlusions remains unknown. We investigated whether oral health was associated with the risk of retinal vascular occlusions. We conducted a retrospective cohort study including 138,484 participants who completed a national health screening program with an oral health examination from the National Health Insurance Service-National Health Screening Cohort (NHIS-HEALS) 2002–2015. Oral health markers, such as the presence of periodontitis, tooth loss, and dental caries, and the frequency of daily tooth brushing, were evaluated. The primary outcome was the occurrence of retinal vascular occlusions up to December 2015. In total, 2533 participants developed retinal vascular occlusions (215 with retinal artery occlusion, 1686 with retinal vein occlusion, 632 with unspecified retinal vascular occlusion). In the multivariable Cox regression analysis, periodontitis was an independent risk factor for retinal vascular occlusions (adjusted hazard ratio: 1.18; 95% confidence interval: 1.02–1.36; *p* = 0.024). Frequent tooth brushing was negatively associated with the risk of retinal vascular occlusions (adjusted hazard ratio: 0.89; 95% confidence interval: 0.80–0.98; *p* = 0.022). Improving oral hygiene may contribute to the attenuation of the risk of retinal vascular occlusions.

## 1. Introduction

Retinal vascular occlusions are a leading cause of visual loss [1,2]. However, the risk factors for retinal vascular occlusion have not been identified. Several studies have shown that systemic atherosclerosis, atrial fibrillation, hypertension, diabetes mellitus, glaucoma, and antiphospholipid syndrome are associated with retinal vascular occlusion, but further research on risk factors or related factors is required [3].

Retinal artery occlusion is mainly caused by atherosclerosis-related complications or embolism, and retinal vein occlusion is related with compression from a neighboring atherosclerotic retinal artery or a prothrombotic stat [1,2]. For the initiation and progression of atherosclerosis and thrombosis, local and systemic inflammation takes a major role [4]. Recent studies have established that poor oral hygiene including periodontitis is associated with inflammatory reactions and thromboembolism [5,6]. Periodontitis, caused by oral bacterial microorganisms, leads to a gradual destruction of periodontal soft tissues and alveolar bone [7]. Periodontitis and dental caries are strongly prevalent diseases producing local inflammatory conditions in surrounding tissues [8]. Insufficient oral care may induce local infections, resulting in chronic inflammation, endothelial dysfunction, and thrombosis, which are important pathological mechanisms contributing to cardiovascular disease [9,10]. There is a complex relationship between periodontitis and retinal vascular occlusion. Systemic inflammation triggered by periodontitis may have adverse effects on cardiovascular risk factors, including diabetes mellitus and atherosclerosis, which may result in retinal vascular occlusion. On the other hand, periodontitis can have a direct impact on retinal vascular occlusion due to inflammatory reactions and thrombosis. To date, no study has been conducted on whether periodontitis has an independent association with retinal vascular occlusion or whether it is a confounding factor. The current study aimed to investigate whether the occurrence of retinal vascular occlusions is linked to oral health markers in a longitudinal study setting.

## 2. Materials and Methods

### 2.1. Data Source

Our study used the National Health Insurance Service-National Health Screening Cohort (NHIS-HEALS) dataset in South Korea [11]. The NHIS biennially provides a complimentary nationwide health screening program for all South Korean adults over 40 years. NHIS-HEALS is a cohort comprising a 10% simple random sample from all health screening participants in 2002–2003 (N = 514,866) [11]. The NHIS-HEALS includes consecutive health screening results, including physical examination, laboratory findings, self-reported survey, and oral examination by professional dentists. The NHIS-HEALS also contains demographic information, death records, socioeconomic status, eligibility, and health claim data (hospital visit, diagnosis code, procedure, and prescription information). The health claim resources are accessible until loss of eligibility for NHIS due to emigration or death of the participants.

### 2.2. Study Population

From the NHIS-HEALS, we identified participants who completed a health screening program, including oral health check-ups, from 2003 to 2004 (baseline examination). Of these, we excluded participants with a history of prior stroke, myocardial infarction, or retinal vascular occlusions before the baseline examination or with missing data for the covariates. A flow chart of participant inclusion and exclusion is shown in Figure 1. The index date of this cohort was defined as the date of baseline examination.

### 2.3. Oral Hygiene Markers and Covariates

At the baseline screening for oral health, the participants were examined for the presence of tooth loss and dental caries by dentists. Periodontitis was identified based on diagnostic codes (International Statistical Classification of Diseases and Related Health Problems 10th Revision, ICD-10 code: K052–K054) input more than twice by a dentist or treatment for periodontal disease (health claim codes: U1010, U1020, U1051–1052, U1071–1072, U1081–U1083) with the diagnostic code of periodontitis for 1 year before the baseline oral health check-up [12,13]. The frequency of daily tooth brushing was dichotomized as ≤1 and ≥2 times per day, based on a self-reported questionnaire in the health check-up program. The presence of dental caries was confirmed by a dentist during a health check-up. The detailed definitions of demographic data (sex, age, smoking status, alcohol consumption, physical activity, household income, body mass index, total cholesterol) and underlying diseases (hypertension, diabetes mellitus, and atrial fibrillation) are demonstrated in Appendix A [14,15,16,17,18].

### 2.4. Study Outcome

The primary outcome was the occurrence of retinal vascular occlusion, which was identified based on the NHIS health claim dataset. Retinal vascular occlusion was defined as the presence of one of the following ICD-10 diagnostic codes with claims for fundoscopic examination (E6660, E6670, E6674) before and after 30 days from the diagnosis: retinal artery occlusion (H34.0, H34.1, H34.2), retinal vein occlusion (H34.8), and unspecified retinal vascular occlusion (H34, H34.9) based on the definition of previous studies [19,20]. After the baseline health examination (index date), all participants were followed up until the development of the primary outcome, loss of eligibility for NHIS due to emigration, death of the participant, or end date of the dataset (31 December 2015).

### 2.5. Statistical Analysis

Data are represented as the proportion of participants (%) or mean ± standard deviation. The characteristic difference between the two groups was compared using the Chi-square test for categorical variables and an independent *t*-test for continuous variables. The cumulative incidence curves according to the markers of oral hygiene were illustrated and compared using a log-rank test. We calculated hazard ratio (HR) and 95% confidence interval (CI) for the oral hygiene markers based on Cox proportional hazard regression models [13]. The proportional hazard assumption in the Cox model was tested based on Schoenfeld residuals, which were satisfied. Multivariable Cox regression models were adjusted for sex, age, smoking status, alcohol consumption, physical activity, household income, body mass index, total cholesterol, and comorbidities [12]. For secondary outcome analysis, individual Cox regression models were constructed for arterial, venous, and unspecified retinal vascular occlusion. Data manipulation and statistical analyses were performed using SAS 9.4 version (SAS Inc., Cary, NC, USA), and the results of two-sided *p*-value < 0.05 were considered significant.

## 3. Results

### 3.1. Baseline Characteristics

According to inclusion and exclusion criteria, we enrolled 138,484 participants who completed a baseline health screening and oral health examination (Figure 1). The mean age at baseline examination was 52.23 ± 8.95 years, and 82,196 (59.35%) participants were male. Among the included 138,484 participants, periodontitis was present in 6.82%. Table 1 shows the characteristics of study participants according to the presence of periodontitis. Tooth loss and frequent tooth brushing (≥2 times per day) were more frequent in participants with periodontitis than those without. The proportion of dental caries was higher in participants without periodontitis than in those with periodontitis.

### 3.2. Risk for the Primary Outcome According to Markers of Oral Hygiene

In the 11.60 ± 1.87 years (mean ± standard deviation) of the follow-up period, 2533 participants developed retinal vascular occlusions (retinal artery occlusion, N = 215; retinal vein occlusion, N = 1686; unspecified retinal vascular occlusion, N = 632). The cumulative incidence curves demonstrated that the occurrence of retinal vascular occlusions increased with the presence of periodontitis, tooth loss, and a lower frequency of daily tooth brushing (Figure 2).

Table 2 shows the results of Cox regression analyses for retinal vascular occlusions. In multivariable Cox regression analyses, the presence of periodontitis was an independent risk factor for the occurrence of retinal vascular occlusions (adjusted HR: 1.18; 95% CI: 1.02–1.36; *p* = 0.024). In contrast, frequent tooth brushing (≥2 times/day) decreased the risk of retinal vascular occlusions (adjusted HR: 0.89; 95% CI: 0.80–0.98; *p* = 0.022). The presence of tooth loss and dental caries were not associated with the occurrence of retinal vascular occlusions. Table 3 shows secondary outcome analysis for arterial, venous, and unspecified retinal vascular occlusions. Frequent tooth brushing (≥2 times per day) was correlated with a decreased risk of unspecified retinal vascular occlusions (adjusted HR: 0.80; 95% CI: 0.66–0.98; *p* = 0.028) compared to tooth brushing ≤ once a day.

## 4. Discussion

Our study found that periodontitis increased the risk of retinal vascular occlusions, while improved oral hygiene behavior such as tooth brushing more than twice a day could lower the risk. Increasing evidence showed that periodontitis is associated with hypertension, diabetes mellitus, cerebrovascular disease, and myocardial infarction [13,21,22]. Self-reported tooth loss due to periodontal disease can increase the risk of venous thromboembolism by 30% [6]. These results imply a possible link between periodontitis and ocular vascular disease. Indeed, a cross-sectional study showed that severe periodontitis was inversely correlated with the arteriovenous ratio, suggesting that periodontitis affects microvascular endothelium function in the retina [23]. A cohort study also demonstrated that patients with periodontitis have a higher risk of developing age-related macular degeneration than those without periodontitis [24]. In US male health professionals, those with periodontal disease had a 1.85-fold higher risk of primary open-angle glaucoma than those without periodontal disease [25]. Similarly, periodontitis was significantly associated with glaucoma (odds ratio: 3.44) in the Korean population [26].

Tooth brushing is a well-established method for removing dental plaque and preventing periodontitis and dental caries [27]. Numerous studies have shown the potential role of tooth brushing in preventing systemic disease [9,13,21,28]. Frequent tooth brushing decreases the risk of new stroke, atrial fibrillation, and heart failure [12,29]. In a health survey among Scottish subjects, participants who reported that they never/rarely brushed their teeth had an increased risk of cardiovascular diseases [30]. Less frequent tooth brushing (≤1 time/day) was also related to obesity and hyperglycemia [28]. In a nationwide cross-sectional study, systolic blood pressure tended to decrease with the tooth brushing frequency [31]. Our study showed an association between retinal vascular occlusion and periodontitis even after adjusting for other risk factors. We added evidence on the association between poor oral health and the occurrence of retinal vascular occlusions in a longitudinal study setting using a nationwide population-based cohort. Our finding highlights that improving oral health, especially with more frequent tooth brushing, may contribute to the prevention of retinal vascular occlusion.

The current study could not provide the exact mechanism underlying the association between oral health markers and retinal vascular occlusions, but the following hypothesis may explain our findings. Periodontitis is a chronic inflammatory disease caused by bacteria that destabilizes the tooth structure and supporting apparatus. The destruction of oral microorganisms can trigger transient or persistent systemic inflammation or bacteremia, leading to extra-oral tissue damage [32]. Inflammatory biomarkers, such as several cytokines released by affected periodontal tissues, can lead to pathological changes in other organs, including retinal vessels [33]. Moreover, the inflammatory response in periodontitis or toxin-mediated bacteremia can produce an immune response, leading to inflammatory vasculopathy and thrombogenesis [34].

Our study found a significant association between tooth loss and retinal vascular occlusions in univariable analysis but not in multivariable analysis. Although dental caries is an important indicator of poor oral hygiene, we could not confirm the association with retinal vascular occlusions, probably since tooth loss and dental caries may have less effect on local inflammatory reactions than periodontitis. Notably, our secondary analysis, based on the classification according to each retinal vascular occlusion, did not reveal a clear association. The discrepancy may be attributed to a lack of information on the disease severity, limited sample sizes, the possible misclassification of diagnoses for retinal vascular occlusion, and residual confounding effects. Further studies are needed to substantiate the association between oral health markers, particularly periodontitis and retinal vascular occlusions. In addition, there were factors more strongly associated with retinal vascular occlusion than oral health parameters. These results suggest that although oral health is important, the control of other related factors should accompany it.

The current study has limitations. First, the dataset only included the Korean population. Second, in our oral health examination dataset, the frequency of tooth brushing was based on a self-reported questionnaire, and thus there may be a recall bias. Third, since consensus on periodontitis was not sufficient in 2003–2004 when the cohort dataset was constructed, it was not able to make agreements about the presence or degree of periodontitis in all participants. Thus, periodontitis was defined in this study using the ICD-10 and treatment claim codes. Fourth, we could not obtain information on the detailed cause of tooth loss. Fifth, marital status, education level, and serum inflammatory markers were not available in the NHIS-HEALS dataset. Sixth, for the measurement of physical activity, days/week was used rather than hours/day, which is less accurate. Finally, although this study has a longitudinal design, the causal relationship between the variables cannot be proved by a retrospective observational study design without interventions.

To summarize, the current cohort study demonstrated that periodontitis showed a positive association, and tooth brushing showed a negative association, with the occurrence of retinal vascular occlusions. Improving oral hygiene may attenuate the risk of retinal vascular occlusions.

## Figures and Tables

**Figure 1 jpm-13-00121-f001:**
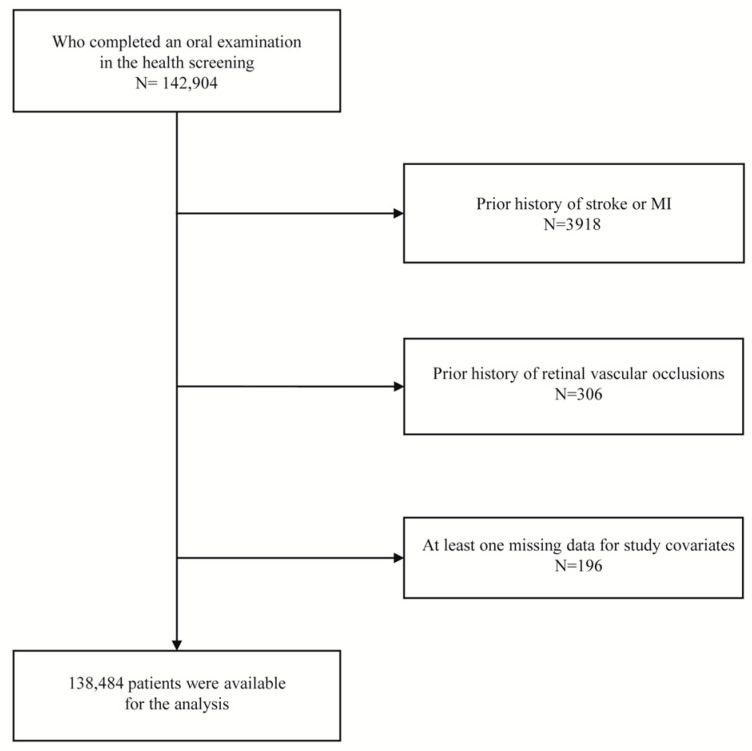
Flow chart of inclusion and exclusion criteria.

**Figure 2 jpm-13-00121-f002:**
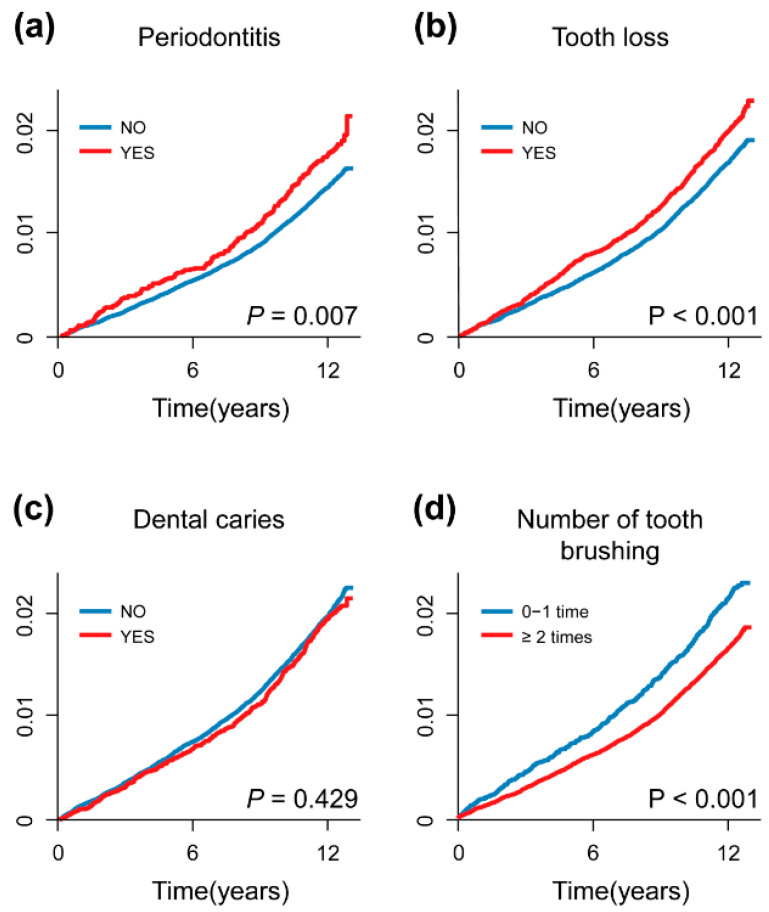
Cumulative incidence curves for the occurrence of retinal vascular occlusions. The occurrence of retinal vascular occlusions increased with the presence of periodontitis, tooth loss, and a lower frequency of daily tooth brushing. X-axis: time (years), Y-axis: cumulative incidence of retinal vascular occlusion.

**Table 1 jpm-13-00121-t001:** Baseline characteristics of the study patients.

Variable	Total N = 138,484	Periodontitis (−) N = 129,043 (93.18%)	Periodontitis (+) N = 9441 (6.82%)	*p*-Value *
Sex, male	82,196 (59.35)	75,916 (58.83)	6280 (66.52)	<0.001
Age, years	52.23 ± 8.95	52.17 ± 8.97	52.93 ± 8.60	<0.001
Smoking status				<0.001
Never smoker	85,364 (61.64)	80,054 (62.04)	5310 (56.24)	
Former smoker	20,118 (14.53)	18,455 (14.30)	1663 (17.61)	
Current smoker	33,002 (23.83)	30,534 (23.66)	2468 (26.14)	
Alcohol consumption, frequency per week				<0.001
<1 time	70,605 (50.98)	65,997 (51.14)	4608 (48.81)	
1–2 times	52,993 (38.27)	49,232 (38.15)	3761 (39.84)	
3–4 times	9804 (7.08)	9065 (7.02)	739 (7.83)	
≥5 times	5082 (3.67)	4749 (3.68)	333 (3.53)	
Physical activity, days per week				<0.001
<1 day	72,040 (52.02)	67,581 (52.37)	4459 (47.23)	
1–4 days	53,079 (38.33)	49,124 (38.07)	3955 (41.89)	
≥5 days	13,365 (9.65)	12,338 (9.56)	1027 (10.88)	
Household income				<0.001
Q1, lowest	35,645 (25.74)	33,678 (26.10)	1967 (20.83)	
Q2,	31,595 (22.81)	29,585 (22.93)	2010 (21.29)	
Q3,	40,870 (29.51)	37,956 (29.41)	2917 (30.87)	
Q4, highest	30,374 (21.93)	27,824 (21.56)	2550 (27.01)	
Body mass index, kg/m^2^	23.92 ± 2.89	23.91 ± 2.90	24.05 ± 2.81	<0.001
Total cholesterol (mmol/L)	5.13 ± 0.94	5.13 ± 0.94	5.16 ± 0.94	<0.001
Comorbidity				
Hypertension	61,017 (44.06)	56,758 (43.98)	4259 (45.11)	0.033
Diabetes mellitus	15,664 (11.31)	14,297 (11.08)	1367 (14.48)	<0.001
Atrial fibrillation	743 (0.54)	690 (0.53)	53 (0.56)	0.732
Oral hygiene markers				
Tooth loss	33,222 (23.99)	30,471 (23.61)	2751 (29.14)	<0.001
Dental caries	26,652 (19.25)	25,427 (19.70)	1225 (12.98)	<0.001
Frequency of daily tooth brushing				<0.001
0–1 time	21,459 (15.50)	20,150 (15.61)	1309 (13.87)	
≥2 times	117,025 (84.50)	108,893 (84.93)	8132 (86.13)	

Data are represented as number of participants (%) or mean ± standard deviation. Q: quartile. * *p*-value is derived from an independent *t*-test or Chi-square test between participant groups with/without periodontitis.

**Table 2 jpm-13-00121-t002:** Risk factors for the occurrence of retinal vascular occlusion.

	Crude HR [95% CI]	*p*-Value	Adjusted HR [95% CI]	*p*-Value
Sex, male	0.79 [0.73–0.86]	<0.001	0.89 [0.80–0.99]	0.026
Age, years	1.05 [1.04–1.06]	<0.001	1.04 [1.03–1.05]	<0.001
Smoking				
Never smoker	1 (Ref)		1 (Ref)	
Former smoker	0.87 [0.77–0.98]	0.017	1.03 [0.90–1.17]	0.694
Current smoker	0.70 [0.64–0.78]	<0.001	0.89 [0.79–1.00]	0.052
Alcohol consumption, frequency per week				
<1 time	1 (Ref)		1 (Ref)	
1–2 times	0.76 [0.70–0.83]	<0.001	0.95 [0.87–1.05]	0.338
3–4 times	0.87 [0.74–1.02]	0.087	1.06 [0.89–1.25]	0.526
≥5 times	0.99 [0.81–1.22]	0.973	0.98 [0.79–1.22]	0.865
Physical activity, days per week				
<1 day	1 (Ref)		1 (Ref)	
1–4 days	0.82 [0.75–0.89]	<0.001	0.98 [0.89–1.07]	0.584
≥5 days	1.11 [0.98–1.26]	0.116	1.02 [0.89–1.15]	0.823
Household income				
Q1, lowest	1 (Ref)		1 (Ref)	
Q2,	0.89 [0.80–0.99]	0.038	1.01 [0.90–1.12]	0.912
Q3,	0.76 [0.69–0.85]	<0.001	0.93 [0.83–1.03]	0.156
Q4, highest	0.78 [0.70–0.87]	<0.001	0.92 [0.82–1.03]	0.145
Body mass index, kg/m^2^	1.05 [1.03–1.06]	<0.001	1.03 [1.02–1.04]	<0.001
Total cholesterol (mmol/L)	1.09 [1.05–1.14]	<0.001	1.02 [0.98–1.06]	0.323
Comorbidity				
Hypertension	1.70 [1.58–1.84]	<0.001	1.33 [1.22–1.44]	<0.001
Diabetes mellitus	1.98 [1.79–2.18]	<0.001	1.63 [1.47–1.80]	<0.001
Atrial fibrillation	1.98 [1.32–2.95]	<0.001	1.36 [0.91–2.03]	0.136
Oral health status				
Periodontitis	1.22 [1.06–1.40]	0.007	1.18 [1.02–1.36]	0.024
Tooth loss	1.18 [1.08–1.29]	<0.001	0.99 [0.91–1.09]	0.891
Dental caries	0.96 [0.87–1.06]	0.434	0.99 [0.89–1.09]	0.828
Frequency of daily tooth brushing				
0–1 time	1 (Ref)		1 (Ref)	
≥2 times	0.78 [0.71–0.86]	<0.001	0.89 [0.80–0.98]	0.022

Data are derived from Cox proportional hazard regression analysis for retinal vascular occlusions. HR: hazard ratio, CI: confidence interval, Q: quartile.

**Table 3 jpm-13-00121-t003:** Secondary outcome analysis for individual outcomes.

	Retinal Artery Occlusion (N = 215)		Retinal Vein Occlusion (N = 1686)		Retinal Vascular Occlusion, Unspecified (N = 632)	
Oral Health Status	Adjusted HR [95% CI]	*p*-Value	Adjusted HR [95% CI]	*p*-Value	Adjusted HR [95% CI]	*p*-Value
Periodontitis	0.98 [0.58–1.66]	0.937	1.19 [0.99–1.42]	0.054	1.23 [0.93–1.62]	0.155
Tooth loss	1.05 [0.77–1.43]	0.762	1.05 [0.94–1.17]	0.403	0.84 [0.69–1.01]	0.065
Dental caries	1.01 [0.72–1.43]	0.947	1.00 [0.89–1.13]	0.983	0.95 [0.77–1.16]	0.595
Frequency of daily tooth brushing						
0–1 time	1 (Ref)		1 (Ref)		1 (Ref)	
≥2 times	0.92 [0.65–1.31]	0.662	0.92 [0.81–1.04]	0.186	0.80 [0.66–0.98]	0.028

Data are derived from Cox proportional hazard regression analysis for each retinal vascular occlusion. HR: hazard ratio, CI: confidence interval. Adjustments were performed for the covariates listed in Table 2.

## Data Availability

The datasets generated and analyzed during the current study are publicly available in the NHIS-HEALS repository created by the National Health Insurance Sharing Service, [NHIS-2022-2-002].

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
