# Peer review of "Oral Health and Risk of Retinal Vascular Occlusions: A Nationwide Cohort Study"

_jpm, 2023, doi:10.3390/jpm13010121_

Round 1

Reviewer 1 Report

The study showed additional evidence about the relationship between oral health and other diseases. This study showed that periodontitis as independent risk factor to retinal vascular occlusion in this national cohort study.

We suggest the author may also explain in the Introduction that the association between periodontitis and retinal vascular occlusion may be more complex and may be confounded by other factors such as systemic diseases since both periodontitis and retinal vascular occlusion influenced by other systemic diseases such as diabetes mellitus.

In materials and methods, the author should explain that the author identified dental caries using ICD 10 too. The author should include the result of blood glucose level of diabetes mellitus since diabetes patient may be examined routinely for blood glucose level or HbA1c level because there was total cholesterol data was available in dataset.

The author may use the duration of activity in hours/day or hours/weeks rather than day/weeks.

In table 1 the mean age total : 52.23 + 8.95, However in line 122 the mean age : 55, 23+ 8.95, was it different data?

In line 123, …138,484 participants, periodontitis was present in 6.82%, was this data available in table 1?

In line 141 figure 2, in the X axis, it was shown time(years), and the number in the figure : 0, 6, 12 in X axis,…was it 6 and 12 years?

in line 186, the author stated that….Our findings are consistent with the previous studies suggesting a relationship between oral hygiene and several systemic diseases, particularly cardi- …did this study showed the association between poor oral health with hypertension, diabetes mellitus and atrial fibrillation?

Reviewer 2 Report

The reviewed manuscript entitled ‘Oral health and risk of retinal vascular occlusions: A nationwide cohort study’ written by Yoonkyung Chang et al. provides interesting associations between occurrence of various types of retinal vascular occlusions and health of oral cavity. The authors evaluated a large population and analyzed many potential confounding factors in multivariate analysis. The article is well structured and scientifically sound. I have only minor comment for this manuscript:

1 1. In Table 2, there are more associated variables than the presence of periodontitis and tooth brushing frequency, and a short comment should be added with respect to these other associations.

I believe that my suggestion will be helpful to the authors to increase the quality of the reviewed manuscript.
